# Affect and post-COVID-19 symptoms in daily life: An exploratory experience sampling study

Gerko Schaap[1]*, Marleen Wensink[1], Carine J. M. Doggen[2,3], Job van der Palen[4,5], Harald E. Vonkeman[1,6], Christina Bode[1]

1 Department of Psychology, Health and Technology, University of Twente, Enschede, The Netherlands, 2 Department of Health Technology and Services Research, Technical Medical Centre, University of Twente, Enschede, The Netherlands, 3 Clinical Research Centre, Rijnstate Hospital, Arnhem, The Netherlands, 4 Department of Epidemiology, Medisch Spectrum Twente, Enschede, The Netherlands, 5 Section of Cognition, Data and Education, University of Twente, Enschede, The Netherlands, 6 Department of Rheumatology and Clinical Immunology, Medisch Spectrum Twente, Enschede, The Netherlands

* g.schaap@utwente.nl

**Data Availability Statement:** The study data and syntaxes are available in the DANS repository: https://doi.org/10.17026/SS/NIDSIJ. The dataset

## Abstract

Insight into the daily life experiences of patients with post-COVID-19 syndrome is lacking. The current study explored temporal fluctuations of and associations between positive and negative affect and symptoms throughout the day in previously hospitalised post-COVID-19 patients using an experience sampling methodology. Ten participants (age: median = 60, interquartile range = 9 years; 50% women; 80% ≥1 comorbidity; 8–12 months since hospital discharge) filled out brief online questionnaires, six times a day for 14 consecutive days. Positive and negative affect, and self-reported symptoms (physical and mental fatigue, cognitive functioning, dyspnoea, and pain) were assessed in real-time. Primarily, graphs were analysed to assess the individual longitudinal courses of and (concurrent and time-lagged) associations between affect and symptoms. Secondly, correlations or multilevel linear regression models were used to support these interpretations. Visual assessment showed limited temporal fluctuation in affect and symptoms. All symptoms appeared to associate positively with each other (correlations between .26 and .85). Positive affect was associated with lower symptoms severity (β's between -.28 and -.67), and negative affect with higher symptoms severity (β's between .24 and .66). Time-lagged analyses showed that–adjusted for residual symptom severity of prior measurements–both types of affect predicted symptom severity two hours later (β's between -.09 and -.31 for positive affect; between .09 and .28 for negative affect). These findings suggest that positive and negative affect may play important roles in post-COVID-19 symptom experience and temporal fluctuation.

## Introduction

While most people recover from a SARS-CoV-2 infection, a substantial group struggles with persistent symptoms. These sequelae are often referred to as long-COVID or as post-COVID-19 syndrome, which can be defined as "signs and symptoms that developed during or after an infection consistent with COVID-19, continue for more than 12 weeks and are not explained

has been fully anonymised and demographic and clinical characteristics (used only for descriptive analysis of the sample) have been obfuscated, as they comprise sensitive data that could compromise patient privacy. The key to these characteristics is available upon reasonable request by contacting the corresponding author.

**Funding:** The author(s) received no specific funding for this work.

**Competing interests:** The authors have declared that no competing interests exist.

by an alternative diagnosis" [1]. Estimated prevalence ranges between 6 to 45 percent of COVID-19 survivors [2–4], depending on definitions, populations, vaccination status, and SARS-CoV-2 variants within these samples. Most frequently reported post-COVID-19 symptoms include fatigue, cognitive dysfunction or brain fog, respiratory problems such as dyspnoea (breathlessness), chronic pain, and loss of sense of smell; often co-occurring [3,5–9]. Previous research is mostly limited to cross-sectional or cohort studies, and few studies looked into the severity of these symptoms, their daily fluctuations, and the interrelations between these symptoms. Similarly, while reviews [5,7] and qualitative studies [10–13] reported the effects on mental health and health-related quality of life, the daily fluctuations over time of positive and negative affect in post-COVID-19 patients have not yet been investigated.

Affect may play an important role in the severity, daily course, and potentially management of post-COVID-19 symptoms. One relevant framework to understand the associations between affect and symptoms is the Broaden-and-Build theory of positive emotions [14,15]. Negative emotions narrow patients' attention and behaviours towards specific action tendencies necessary for survival, such as fear evoking an urge to escape [16]. In contrast, positive emotions can broaden patients' perspectives and approaches and help build personal physical, cognitive, social and psychological resources [14]. For health, this translates into the risk for negative affect to restrict one's attention on symptoms, whereas positive affect may help patients to find additional resources to cope with their symptoms and aid with self-management in general. This relationship between affect and symptoms has been found in many patient populations, and the directions of these relationships can be surprising; while it seems common sense that negative affect follows from increased symptom severity, the reverse can also be true. Both negative and positive affect have been found to be bidirectionally associated with or predictive of chronic symptoms such as fatigue, dyspnoea, and pain in various patient populations [17–22]. These studies mostly made use of cross-sectional designs and investigated these relations at a trait level or relatively stable level, and hence provide little insight into short-term associations. In addition, investigating whether affect informs an increase or decrease of symptom severity shortly later allows for a richer, more detailed and nuanced understanding of illness experiences on a daily level. Moreover, this understanding may guide in identifying opportunities for more advanced interventions; advanced in the sense of potential for tailoring and personalisation. Currently it is not known whether positive and negative affect are predictive of post-COVID-19 symptoms, and how these are experienced real-time over the day.

To address the gaps in current knowledge, this observational study aimed to explore the daily courses of and interrelations between positive and negative affect and post-COVID-19 symptom severity, such as of fatigue, cognitive dysfunction, dyspnoea, and pain. An Experience Sampling Methodology (ESM) has been applied, which allowed investigating these aspects in daily life through repeated real-time assessments. Moreover, this method helps overcome recall bias, and due to the repeated observations within participants, allows for rich and robust insights in patients' daily experiences, even in small patient samples [23]. The following research questions were to assess: (1) temporal fluctuations of affect and post-COVID-19 symptoms; (2) whether these symptoms were associated with (a) positive and (b) negative affect measured concurrently; and (3) whether symptoms were predicted by (a) positive and (b) negative affect measured at an earlier timepoint using time-lagged analyses.

## Methods

### Recruitment

This study is part of a larger, on-going cohort study on the long-term impact of severe COVID-19 requiring hospitalisation (registered at ClinicalTrials.gov; NCT05813574). For the

present study, participants were selected based on self-reported health status three months after hospital discharge and their self-assessed health status at the time of recruitment. Formerly hospitalised COVID-19 patients who indicated their health to be much worse compared to a year ago on the Dutch Short Form Health Survey (SF-36) [24] were selected for an interview [13]. During the interview, approximately 9 months after hospital discharge, self-perceived health status was reassessed, and participants were screened for self-reported symptoms (e.g. fatigue). If participants still suffered from post-COVID-19-associated symptoms, they were invited for the present study. Out of the 16 selected candidates, 11 were willing to participate.

Inclusion criteria were: a) having been discharged from hospital after polymerase chain reaction-confirmed acute COVID-19; b) persisting, fluctuating symptoms attributed by the participant to post-COVID-19 syndrome; c) ≥18 years of age; d) proficiency in Dutch; and e) willingness to install a smartphone-based measurement app. Participants were excluded from analysis in case of compliance rates <30% of the daily assessments [23]. All participants provided written informed consent, and the Medisch Spectrum Twente Institutional Review Board (K20-30) and the Ethics Committee Behavioural, Management and Social Sciences of the University of Twente, the Netherlands (210799) approved the study.

## Procedure and measurements

Using an ESM setup, this study set out to examine daily fluctuations of and associations between affect and self-reported post-COVID-19 symptom severity. Data were collected between September 5 and October 27, 2021, using the Ethica platform (ethicadata.com) and the related Ethica smartphone application. ESM data consisted of daily questionnaires on symptoms, feelings, and behaviours. Additional data were collected, but were beyond the scope of this study and are discussed elsewhere [25]. See S1 File for the complete questionnaires in Dutch (original) and English (translated). Demographic and health-related characteristics were retrieved from hospital medical records with participants' approval.

Before the data collection started, participants were carefully instructed on how to install and set up the application, on how the questionnaires worked, and on what was expected from them (i.e. to respond to as many questionnaires as possible). Participants could practice with the questionnaire before the study started. Finally, participants were informed to contact the researchers if and when necessary before the start of and during the data collection period.

Participants were followed for 14 consecutive days via seven daily self-report questionnaires prompted by a beep signal, of which six were relevant for this study. These short questionnaires–taking approximately two minutes–were prompted semi-randomly within two-hour intervals at 08.00–09.59, 10.00–11.59, 12.00–13.59, 14.00–15.59, 16.00–17.59, and 18.00–19.59 (i.e. signal-contingent sampling). Each notification expired after 15 minutes to prevent reactivity (i.e. participants 'preparing themselves' for the prompts) without being too inconvenient [23,26].

The questionnaires assessed, in fixed order, symptom severity and affect. Fatigue–assessed as physical fatigue and mental fatigue–and dyspnoea were observed real-time using statements ("right now, I feel shortness of breath") that could be answered using 7-point Likert-type answers: 1 –'strongly disagree' to 7 –'strongly agree'. Cognitive function was assessed by asking for clarity of thought ("how is your 'thinking' (concentration, memory, attention) right now?") on a visual analogue scale (VAS): 0 –'slow and difficult' to 10 –'sharp and alert'. This scale was reversed for analysis (cognitive dysfunction), so that higher scores indicated higher severity. Pain was assessed first by asking if the participant experienced pain 'right now' (presence of pain). If affirmative, four additional prompts opened up to assess the severity of headache,

joint pain, chest pain, and pain 'elsewhere in the body' or non-specific, which could be answered on four VAS: 0 –'no pain' to 10 –'worst imaginable pain'. However, as these pains varied too much between participants, the VAS were omitted from further analysis. Nevertheless, their pain courses over time per participant were visualised and are reported in S4 File. Affect was assessed using statements on five negative (anxious, gloomy, sad, irritable, and disappointed) and five positive feelings (excited, relaxed, satisfied, thankful, and joyful; e.g. "right now, I feel satisfied"), to be answered on a 7-point Likert-type scale: 1 –'strongly disagree' to 7 –'strongly agree'. Two mean scales of the five positive affect items and five negative affect items were created as intercorrelations within these constructs were very high (both: Cronbach's α = .96). Additionally, a categorical variable (affect category) was made to indicate whether participants were predominantly positive (positive affect >4), predominantly negative (negative affect >4) or neither predominantly positive nor negative ('neutral'). All items were based on previously validated questionnaires or ESM studies [27–34].

## Data analysis

ESM data consisted of observations (level 1) within days (level 2) within individuals (level 3). To make full use of the richness of these observations, data were primarily assessed visually. Observations were plotted for exploration per participant (affect and symptoms), per symptom by affect category, and per symptom at the current time (t) with affect category of the previous time point (lagged t-1). The graphs were plotted in RStudio version 23.03.0 +386 [35].

Additionally, statistical approaches were used to support these interpretations. Statistical evidence for associations between symptoms in general were explored using personal averages (Person Mean) of the severity of each symptom within each participant. Spearman rank correlations ($r_s$) were used to account for non-parametric distributions and the small sample size.

To assess potential associations between affect and symptoms structured as two levels (observations nested within participants), multilevel linear regression (MLR) models via the SPSS MIXED procedure were used. These models, also referred to as hierarchical linear models, are appropriate and often used for ESM data and account for randomly missing observations, which is common in ESM studies [23,26]. The models accounted for participants (*subjects*) and timepoints (*repeated measures*), and were set up with restricted maximum likelihood estimations and first-order autoregressive (AR1) covariance structure, which was found to be best fitting according to the Akaike Information Criterion (AIC). As variables were measured with different scales, Z-scores were used for analysis and to retrieve standardised regression estimates. In the concurrent MLR models, each symptom was the outcome variable with either positive affect or negative affect as the fixed covariate. Due to a strong correlation between positive affect and negative affect ($r_s$ = -.82), these variables were examined separately in univariate MLR. To examine the prospective associations, positive and negative affect were lagged to the previous timepoint (t-1) in relation to the timepoint of the symptom. For these models, each first symptom measurement of the day was omitted, as the time between prior affect and symptom was too long (e.g. affect at T6 (e.g. 18.30) was adjusted to not predict an effect on fatigue next day at T7 (08.30)). Additionally, the outcome symptom was lagged t-1 and added as fixed covariate, to adjust for a continuance or residual experience of the symptom. Statistical significance was assessed using 95% confidence intervals and α < .05. Standardised regression estimates were interpreted as small (>.1), medium (>.3) or large (>.5) effect size following Cohen's guidelines [36]. Descriptive statistics are presented in absolute number and percentages or median (*Mdn*) and interquartile range *(IQR)*. IBM SPSS Statistics 28 was used for all statistical analyses.

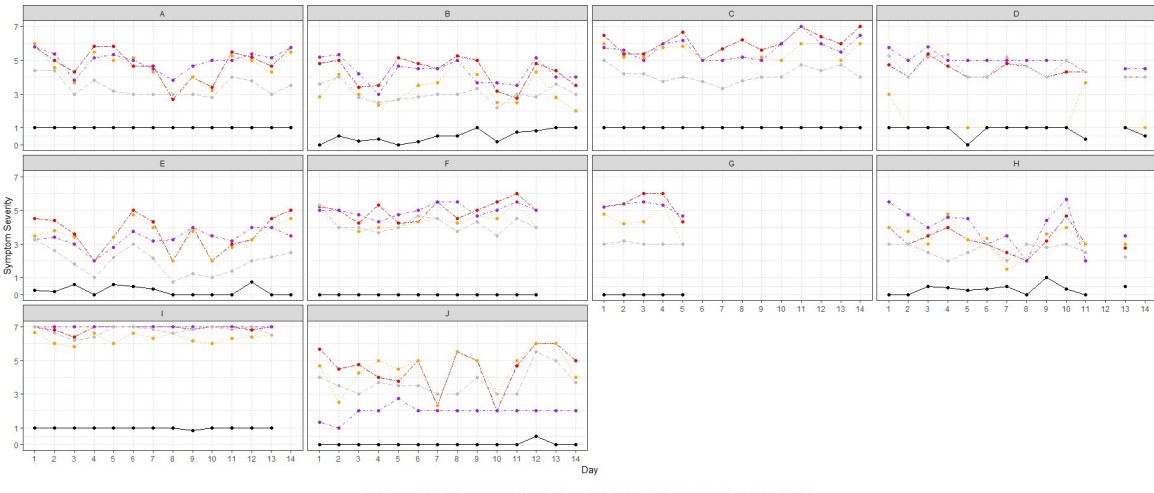

**Fig 1.** Daily average symptom severity over 14 days for participants A–J. Pain presence was a dichotomous variable, with 1 = present and 0 = absent. Consequently, scores of e.g. 0.8 indicate that a participant did mostly but not always experience pain that day. See Fig 2 and S2 File for more detailed insights in symptom courses.

## Results

### Sample characteristics

After removing one participant with a compliance rate <30%, the final sample comprised 10 participants, who were between 48 and 76 (*Mdn* = 60, *IQR* = 9) years old and half of whom were women (*n* = 5). Three participants had one comorbidity, three participants had two or more and four had no comorbidities. Reported comorbidities were chronic respiratory disease (*n* = 2), gastrointestinal disease (*n* = 2), hypertension (*n* = 2), hypothyroidism (*n* = 1), cancer survivor (*n* = 1), cardiovascular (*n* = 1), diabetes mellites (*n* = 1), rheumatic disease (*n* = 1). All but two participants (80%) were overweight (Body Mass Index ≥25; *Mdn* = 29.0, *IQR* = 14.6, *Min* = 22.0, *Max* = 41.8). Observations were made between 8 and 12 (*Mdn* = 10, *IQR* = 2) months after hospital discharge. Compliance rates of participants varied between 32 and 92% (*Mdn* = 63.5%, *IQR* = 40.1%), with in total 502 completed questionnaires.

### Courses of post-COVID symptoms and affect

Visual assessment of the daily average symptom severity per participant (Fig 1) suggested that there were clear between-person differences in experiences of symptoms. Furthermore, physical and mental fatigue, dyspnoea and, to a lesser extent, cognitive dysfunction associated with each other over time on a daily level: an increase in one symptom co-occurred with an increase in the other. This interpretation was supported by correlational analyses of person mean severities ($r_s$ between .37 and .85, $p$ < .001; Table 1). For example, if a participant tends to experience physical fatigue, they also tend to experience cognitive dysfunction ($r_s(8) = .80$, $p$ < .001). Although less obvious from the visual assessment, presence of pain likewise was weakly to moderately correlated to higher severity of other symptoms ($r_s$ between .26 and .54, $p$ < .001; Table 1). Visual assessment of the graphs (see Fig 2 for the course of physical fatigue per participant; see S2 File for courses of mental fatigue, cognitive dysfunction, dyspnoea and pain per participant) revealed that symptoms fluctuated over time to various degrees. Cognitive dysfunction and pain showed the least variability and were mostly stable within participants. Aside from that, no clear temporal patterns were observed within participants. To illustrate,

Table 1. Spearman correlations between symptoms at person mean level.

|  | Physical fatigue | Mental fatigue | Cognitive dysfunction | Dyspnoea |
|---|---|---|---|---|
| **Physical fatigue** | - |  |  |  |
| **Mental fatigue** | .76*** | - |  |  |
| **Cognitive dysfunction** | .73*** | .37*** | - |  |
| **Dyspnoea** | .88*** | .53*** | .66*** | - |
| **Pain presence** | .25*** | .32*** | .27*** | .37*** |

*** = $p < .001$.

while physical fatigue often was lower in the afternoon for participant B, this was not necessarily the case every day (Fig 2).

Affect was mostly stable within participants, with two participants (C and I, Fig 2; see S4 File for courses of and interrelations between positive and negative affect per participant) predominantly experiencing negative affect. No evidence for emotional instability was found (S4 File).

## Associations between affect and post-COVID symptoms

**Concurrent associations.** Overall, negative affect co-occurred with higher severity in all symptoms (Fig 2 and S2 File). Additionally, experiencing neither predominantly positive nor negative ('neutral') affect co-occurred with higher symptom severity. Conversely, predominantly positive affect co-occurred more often with lower symptom severity. Pain seemed to be present regardless of affective tendencies in most participants. MLR models (Table 2) supported these assessments for all associations, and seem to suggest that both affects influence symptoms to a similar extent. For example, cognitive dysfunction was (almost) strongly and significantly associated negatively with positive affect (ß = -.46) and positively with negative affect (ß = .51).

**Time-lagged associations.** To explore whether symptoms were influenced by affect, positive and negative affect were lagged to the prior observation (t-1; visualisations in S3 File). As

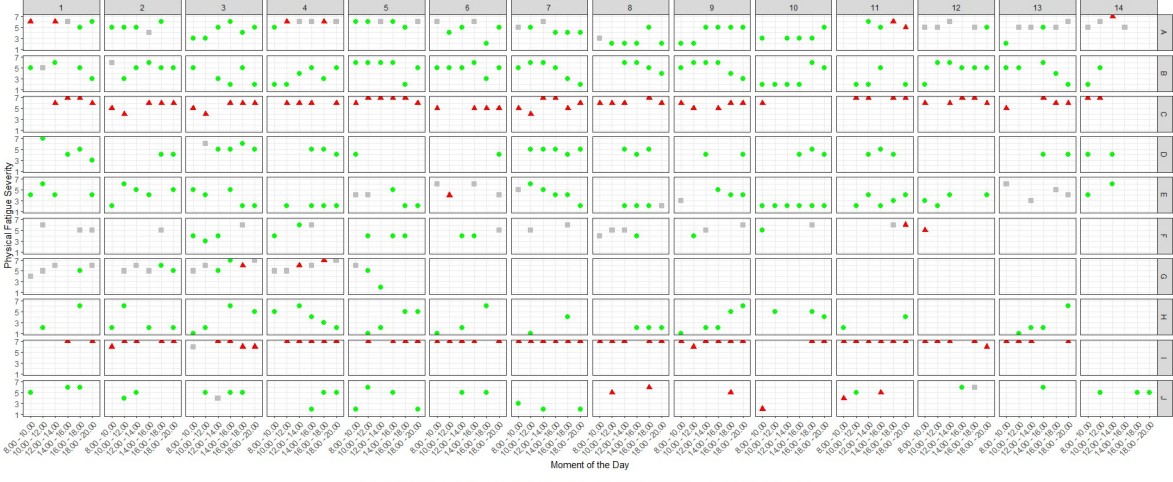

**Fig 2.** Course of physical fatigue over 14 days with corresponding affect per participant A–J.

**Table 2. Overview of 10 univariate models of positive and negative affect as concurrent predictors of symptoms.**

| Outcome | Positive affect | | | Negative affect | | |
|---|---|---|---|---|---|---|
| | ß | SE | 95%CI | ß | SE | 95%CI |
| Physical fatigue | -.67 | 0.04 | -.75 to -.59 | .59 | 0.05 | .49 to .69 |
| Mental fatigue | -.59 | 0.05 | -.69 to -.49 | .66 | 0.05 | .56 to .76 |
| Cognitive dysfunction | -.46 | 0.05 | -.55 to -.36 | .51 | 0.06 | .39 to .62 |
| Dyspnoea | -.49 | 0.05 | -.60 to -.39 | .37 | 0.07 | .24 to .50 |
| Pain presence | -.28 | 0.06 | -.39 to -.16 | .24 | 0.07 | .10 to .38 |

Standardised estimates of fixed covariate (either positive or negative affect) on outcome (symptom) in univariate multilevel linear regression models. All associations were statistically significant, $p < .001$. SE = standard error; 95%CI = 95% confidence intervals presented with respectively lower and upper limits.

affect was relatively stable within participants, influences are inconclusive, but a change from positive to 'neutral' or negative affect might indicate a worsening of symptom severity (see e.g. Participant A). MLR models showed that although symptom severity often was a continuance or remnant of previous measurement–especially for the more stable symptoms such as cognitive dysfunction and presence of pain–both positive and negative affect still were significant but mostly weak predictors (Table 3), suggesting a small influence of affect on symptoms two hours later. Post-hoc analyses showed that, in line with the stable affect trajectories, time-lagged symptoms were not significant predictors of positive and negative affect two hours later.

## Discussion

This study explored the daily courses of and interrelations between affect and post-COVID-19 symptoms using an ESM approach. The main finding was that both positive and negative affect predicted symptom severity both concurrently and two hours later. Moreover, post-COVID-19 symptoms were found to fluctuate in severity both within and between days, and these severities fluctuated in co-occurrence with each other.

This study showed that affect might play a role in the experience of post-COVID-19 symptom severity. This finding is in line with expectations that symptom severity is reduced in

**Table 3. Overview of 10 multivariate models of positive and negative affect as time-lagged predictors of symptoms.**

| Outcome | Predictor | Positive affect | | | Negative affect | | |
|---|---|---|---|---|---|---|---|
| | | ß | SE | 95%CI | ß | SE | 95%CI |
| Physical fatigue | Affect t-1 | -.31 | 0.05 | -.41 to -.20 | .27 | 0.05 | .17 to .36 |
| | Physical fatigue t-1 | .46 | 0.05 | .36 to .57 | .51 | 0.05 | .41 to .60 |
| Mental fatigue | Affect t-1 | -.18 | 0.05 | -.27 to -.08 | .28 | 0.05 | .19 to .38 |
| | Mental fatigue t-1 | .65 | 0.05 | .56 to .74 | .54 | 0.05 | .45 to .64 |
| Cognitive dysfunction | Affect t-1 | -.14 | 0.04 | -.21 to -.07 | .12 | 0.03 | .06 to .19 |
| | Cognitive dysfunction t-1 | .82 | 0.04 | .75 to .89 | .85 | 0.03 | .78 to .91 |
| Dyspnoea | Affect t-1 | -.22 | 0.04 | -.31 to -.14 | .18 | 0.04 | .10 to .26 |
| | Shortness of t-1 | .61 | 0.05 | .52 to .71 | .67 | 0.04 | .58 to .75 |
| Pain presence | Affect t-1 | -.09[†] | 0.03 | -.15 to -.03 | .09[†] | 0.03 | .03 to .15 |
| | Pain t-1 | .82 | 0.03 | .76 to .89 | .83 | 0.03 | .76 to .90 |

Standardised estimates of fixed covariates (residual symptom and (either positive or negative) affect) on outcome (symptom) in multivariate multilevel linear regression models. The first preceding measurement of each predictor is represented as t-1. All associations were statistically significant at $p < .001$, except for [†]: $p = .006$.
SE = standard error; 95%CI = 95% confidence intervals presented with respectively lower and upper limits.

occurrence with and after positive affect, which is based on the Broaden-and-Build framework [14]. Moreover, the results are in accordance with previous findings on the relationships between trait affect and fatigue, dyspnoea and pain in other patient populations [17,20,21,37]. Furthermore, they are comparable to findings in functional somatic syndromes such as fibromyalgia and chronic fatigue syndrome. For these syndromes, it has been reported that negative affective states and negative trait affect elicit increased symptom severity [22,38], and that positive trait affect was associated with decreased symptom severity [22]. The present study suggests that similar affect–symptom relationships also exist in post-COVID-19 patients, and adds to the understanding that these relationships occur on a momentary level and have short-term persisting effects by showing that small to medium effects of positive and negative affect on symptoms can also be observed two hours later.

Based on visual assessment, a difference in the relationship of positive and negative affect with symptom severity was observed. Negative affect was mostly consistent within two participants, while positive affect fluctuated more often and co-occurred with fluctuating symptom severity. This might suggest that there are important between- and within-individual differences. Previous findings in a general population showed that relationships between affect and somatic symptoms are better explained on a between-individual level by negative affect and on a within-individual level by positive affect [39]. Consequently, future studies on the relationships between affect or psychological issues (such as depression, anxiety or uncertainty) and post-COVID-19 symptoms should account for within-individual differences that may be obscured by between-individual findings. Moreover, for treatment, this indicates that different (affect-related) psychological approaches might be necessary and promising when dealing with symptom (self-)management.

Results also showed between- and within-individual differences in symptom severity. Not all participants experienced the same symptoms, and did not experience these symptoms at similar times. Moreover, no clear temporal trajectories could be observed in patients who experienced certain symptoms. For example, participants either experienced always severe or no dyspnoea, or the severity fluctuated for–as of yet–undetermined reasons (S2 File). Aside from small influences of affect, these fluctuations may be partially explained by exertion and daily activities, as previously explored for physical and mental fatigue in this sample [25].

Overall, present symptoms seemed to mostly co-occur, as they associated positively with each other over time within this sample–especially within somatic symptoms. Previously, it was shown using cross-sectional measures that many post-COVID-19 patients experience multiple symptoms (e.g. combinations of fatigue and pain, respiratory issues, and cognitive problems) at the same time, adding to the experienced disease burden [3]. The current study adds to the understanding that symptoms might influence each other's severity. Interestingly, these associations could also be beneficial for patients, as it suggests that successfully targeting one symptom (e.g. dyspnoea) for an intervention might also result in a reduction in other symptoms (e.g. physical fatigue and pain). Although the present study cannot provide full evidence for this, these general trends may indicate that in accordance with the Broaden-and-Build framework and previous empirical findings that investigated sum score measures of symptoms [e.g., 39,40], an overall increased well-being might similarly result in an overall decrease in symptom severity.

Taken together, our findings, especially on the role of positive affect, may serve as ground for exploration of an affect-based intervention, such as a positive psychological intervention (PPI). PPI have been found to be effective especially in chronic pain management and self-regulation [41,42], and in reducing distress [43]. Therefore, they seem useful for post-COVID-19 symptom management. Additionally, an advantage of PPI is that they may provide participants with a sense of control over their rehabilitation. Perceived control was found to be

severely impacted in post-COVID-19 patients [13]. Moreover, the effects of PPI are quickly noticeable by participants [see e.g. 44]. To make best use of these interventions, tailoring to participants' needs might be necessary. Given the temporal association between (positive) affect and symptom severity, tailored adaptive interventions that identify specific times at which PPI could best take place (e.g. just-in-time adaptive interventions) seem promising. Future studies should explore the best targets for these times, or whether these interventions should even be time-based.

The strength of this study is the experience sampling methodology, which allowed for an in-depth exploration of real-time assessed affect and symptom severity, increasing the reliability and ecological validity [23]. Moreover, by plotting participants' individual experiences of affect and symptoms over the day, a deep understanding was gained. Although this analytical strategy compensated for the small sample, the sample size can be seen as a limitation. Care should be taken to generalise the results. Evidence for temporal symptom patterns and stronger affect variability or instability may have arisen in a larger sample. Moreover, while the number of observations included was high, an average response rate of 60% is–while acceptable for ESM studies [23,26]–rather low. Additionally, the sample was mostly homogeneous and consisted of patients with a relatively long course of post-COVID-19 syndrome following COVID-19 hospitalisation. Participants may have already learnt to live with their illness, as well as how to deal with certain negative emotions. Consequently, more recently diagnosed patients may experience more negative affect or higher symptom severity. Similarly, the effects of (the number of) vaccinations have not been accounted for. These have been shown to produce certain ameliorating effects on post-COVID-19 syndrome [45]. Finally, a selection of post-COVID-19-related symptoms was made based on prior research [5–9,13]. In-depth information on all (self-)reported signs and symptoms attributed to post-COVID-19 syndrome were not collected as this was deemed to be beyond the scope of our study (e.g. symptoms without expected daily fluctuation, such as anosmia or persistent fever) and too obtrusive for participants.

## Conclusions

This study used an Experience Sampling Methodology to assess the temporal courses of and relations between affect and symptoms in the post-COVID-19 patients. It provided important insights in the momentary and prospective interactions between affect and post-COVID-19 symptoms. Both positive and negative affect play small but important roles in the severity of fatigue, cognitive dysfunction and dyspnoea and the presence of pain. Future studies should replicate and further explore these associations in large and more diverse samples, to use insights for advanced psychological interventions for post-COVID-19 patients.

## Supporting information

**S1 File. Morning and momentary questionnaires.**
(PDF)

**S2 File. Visualisations of courses of symptoms with concurrent affect.**
(PDF)

**S3 File. Visualisations of courses of symptoms with time-lagged affect.**
(PDF)

**S4 File. Visualisations of courses of affect and of specified pains.**
(PDF)

## Acknowledgments

The authors thank Peter ten Klooster for his advice and support in data management and the statistical analyses.

## Author Contributions

**Conceptualization:** Gerko Schaap, Carine J. M. Doggen, Job van der Palen, Harald E. Vonkeman, Christina Bode.

**Data curation:** Gerko Schaap, Marleen Wensink.

**Formal analysis:** Gerko Schaap.

**Funding acquisition:** Carine J. M. Doggen, Job van der Palen, Harald E. Vonkeman, Christina Bode.

**Investigation:** Gerko Schaap, Marleen Wensink.

**Methodology:** Gerko Schaap, Marleen Wensink, Harald E. Vonkeman, Christina Bode.

**Supervision:** Harald E. Vonkeman, Christina Bode.

**Visualization:** Gerko Schaap.

**Writing – original draft:** Gerko Schaap.

**Writing – review & editing:** Gerko Schaap, Marleen Wensink, Carine J. M. Doggen, Job van der Palen, Harald E. Vonkeman, Christina Bode.

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
