## [Decision Letter · Decision Letter 0]

4 Jan 2024

PONE-D-23-37972Affect and post-COVID-19 symptoms in daily life: An exploratory experience sampling studyPLOS ONE

Dear Dr. Schaap,

Thank you for submitting your manuscript to PLOS ONE. After careful consideration, we feel that it has merit but does not fully meet PLOS ONE’s publication criteria as it currently stands. Therefore, we invite you to submit a revised version of the manuscript that addresses the points raised during the review process.

We look forward to receiving your revised manuscript.

Kind regards,

Kamal Sharma

Academic Editor

PLOS ONE

Journal Requirements:

3. In the online submission form, you indicated that [Data cannot be shared publicly because of containing pseudonymised personal data. Data are available upon reasonable request for researchers who meet the criteria for access to confidential data by contacting the corresponding author. The syntaxes used for data analysis will be available on

DANS: https://doi.org/10.17026/SS/NIDSIJ]. 

Additional Editor Comments:

Hello, thanks for choosing the journal for your research.

The current draft needs minor revisions as pointed out by our reviewer.

Please do the corrections and revert with the same.

Thanks

Reviewers' comments:

Reviewer's Responses to Questions

**Comments to the Author**

1. Is the manuscript technically sound, and do the data support the conclusions?

Reviewer #1: Yes

2. Has the statistical analysis been performed appropriately and rigorously? 

Reviewer #1: Yes

3. Have the authors made all data underlying the findings in their manuscript fully available?

Reviewer #1: Yes

4. Is the manuscript presented in an intelligible fashion and written in standard English?

Reviewer #1: Yes

5. Review Comments to the Author

Reviewer #1: The authors have attempted to study one of the interesting concepts of positive and negative affect and its association with post COVID symptoms. Following suggestions shall be considered

1)As per author’s description, it seems that Post-COVID symptom diagnosis is patient perceived (no clarity on clinician’s involvement in the same), so it shall be explicitly mentioned as “Self-reported post COVID symptoms”.

2)Did patients were oriented towards the questionnaire/ESM? It is important to mention in methodology section.

3) Also, as post COVID symptom is a cluster term imbibing numerous conditions, a detailed information regarding type of post COVID symptom and clinical condition of the patients, type and duration of comorbidity, management of comorbidity needs to be provided in the result section.

After careful addressal of the above-mentioned points, the manuscript can be considered for publication.

6. PLOS authors have the option to publish the peer review history of their article (what does this mean?). If published, this will include your full peer review and any attached files.

Reviewer #1: **Yes: **Komal Shah

---

## [Author Response · Author response to Decision Letter 0]

14 Feb 2024

Thank you for your time and for inviting us to revise our manuscript, Affect and post-COVID-19 symptoms in daily life: An exploratory experience sampling study. We have outlined below our responses to the comments by the reviewer and the academic editor. 

Responses to the academic editor

Journal Requirements:

Response: Thank you for this comment. We have reviewed the style requirements and confirm that the manuscript should be in line.

3. In the online submission form, you indicated that [Data cannot be shared publicly because of containing pseudonymised personal data. Data are available upon reasonable request for researchers who meet the criteria for access to confidential data by contacting the corresponding author. The syntaxes used for data analysis will be available on

DANS: https://doi.org/10.17026/SS/NIDSIJ]. 

Response: Thank you for pointing this out. We are aware of the benefits of open science and publicly available data, and fully agree with you. We have managed to anonymise our data in line with guidelines and restrictions of both Dutch legal and hospital (ethical) policies. The dataset has now been uploaded to DANS along with the syntaxes. Our data availability statement has been updated as follows:

“The study data and syntaxes are available in the DANS repository: https://doi.org/10.17026/SS/NIDSIJ. The dataset has been fully anonymised and demographic and clinical characteristics (used only for descriptive analysis of the sample) have been obfuscated, as they comprise sensitive data that could compromise patient privacy. The key to these characteristics is available upon reasonable request by contacting the corresponding author.”

Response: The references have been checked to ensure that all citations are correct, refer to the most recent version, and have not been retracted. One citation has been altered to refer to a more recent version (Lopez-Leon et al., 2021). Additionally, three citations have been slightly revised to better fit the journal reference style requirements. 

Responses to the reviewer

Reviewer #1: The authors have attempted to study one of the interesting concepts of positive and negative affect and its association with post COVID symptoms. Following suggestions shall be considered

We thank the reviewer for the thoughtful and constructive feedback, which helped us improve our manuscript. 

1)As per author’s description, it seems that Post-COVID symptom diagnosis is patient perceived (no clarity on clinician’s involvement in the same), so it shall be explicitly mentioned as “Self-reported post COVID symptoms”.

Response: Post-COVID-19-syndrome was indeed self-assessed (as described under Recruitment), and refers to self-reported symptoms that were attributed by the participants to this syndrome and not attributable to health conditions prior to COVID-19 and hospitalisation. Throughout the manuscript, we have added ‘self-reported’ where this might have led to confusion, or have ensured that this should be clear to readers. 

2)Did patients were oriented towards the questionnaire/ESM? It is important to mention in methodology section.

Response: We agree we you that this is important information. Participants were instructed and informed on the nature of ESM in this study, as well as the use of the application. We have added a paragraph explaining this in the Procedure section:

“Before the data collection started, participants were carefully instructed on how to install and set up the application, on how the questionnaires worked, and on what was expected from them (i.e. to respond to as many questionnaires as possible). Participants could practice with the questionnaire before the study started. Finally, participants were informed to contact the researchers if and when necessary before the start of and during the data collection period.” 

3) Also, as post COVID symptom is a cluster term imbibing numerous conditions, a detailed information regarding type of post COVID symptom and clinical condition of the patients, type and duration of comorbidity, management of comorbidity needs to be provided in the result section.

Response: Thank you for this comment. We have not collected in-depth information on all (self-)reported signs and symptoms attributed to post-COVID-19 syndrome, as this was beyond the scope of our study. The symptoms of interest in our study (physical and mental fatigue, dyspnoea, cognitive dysfunction and pain, as well as negative affect) were based on prior research (Schaap et al., 2022) in which no other post-COVID-19-related symptoms were described as being present at that time by our participants. As described in our introduction, findings in meta-analyses supports the decision to focus on the selected symptoms. 

Furthermore, participants were selected on self-assessment of post-COVID-19 syndrome, including self-attribution of the symptoms to long-term sequelae of COVID-19. As the influence of (some) comorbidities on post-COVID-19 syndrome is yet unclear, we have reported these. In our overarching project (see Clinical Trials NCT05813574), we will investigate the potential influence of comorbidities in COVID-19 survivors, but this falls beyond the scope of the current study. We have now added the comorbidities in the Results section:

“Reported comorbidities were chronic respiratory condition (n = 2), gastrointestinal disease (n = 2), hypertension (n = 2), hypothyroidism (n = 1), cancer survivor (n = 1), cardiovascular (n = 1), diabetes mellites (n = 1), rheumatic disease (n = 1).“

---

## [Decision Letter · Decision Letter 1]

3 May 2024

PONE-D-23-37972R1Affect and post-COVID-19 symptoms in daily life: An exploratory experience sampling studyPLOS ONE

Dear Dr. Schaap,

Thank you for submitting your manuscript to PLOS ONE. After careful consideration, we feel that it has merit but does not fully meet PLOS ONE’s publication criteria as it currently stands. Therefore, we invite you to submit a revised version of the manuscript that addresses the points raised during the review process.

We look forward to receiving your revised manuscript.

Kind regards,

G. K. Balasubramani

Academic Editor

PLOS ONE

Journal Requirements:

Reviewers' comments:

Reviewer's Responses to Questions

**Comments to the Author**

1. If the authors have adequately addressed your comments raised in a previous round of review and you feel that this manuscript is now acceptable for publication, you may indicate that here to bypass the “Comments to the Author” section, enter your conflict of interest statement in the “Confidential to Editor” section, and submit your "Accept" recommendation.

Reviewer #2: (No Response)

Reviewer #3: All comments have been addressed

2. Is the manuscript technically sound, and do the data support the conclusions?

Reviewer #2: Partly

Reviewer #3: Yes

3. Has the statistical analysis been performed appropriately and rigorously? 

Reviewer #2: No

Reviewer #3: Yes

4. Have the authors made all data underlying the findings in their manuscript fully available?

Reviewer #2: Yes

Reviewer #3: Yes

5. Is the manuscript presented in an intelligible fashion and written in standard English?

Reviewer #2: Yes

Reviewer #3: Yes

6. Review Comments to the Author

Reviewer #2: Numerous publications have extensively covered this topic (many on PlosONE too). It is well-established that these symptoms exhibit correlations. However, it seems unlikely that this manuscript will contribute novel insights to the existing research.

The sample size is notably limited, rendering the results unreliable for any statistical model. It may be appropriate to perform descriptive analysis only. A general guideline recommends having one covariate per 10 subjects.

It's worth noting that the Spearman correlation does not account for within-subject correlation over time; it may not be appropriate to combine timepoints together.

In Figure 1, the pain curves display limited variation, making it challenging to believe in significant correlations with other symptoms.

Reviewer #3: The authors investigated the daily courses and interrelations between positive and negative affect and post-COVID-19 symptom severity by using an ESM approach. The authors demonstrated that post-COVID-19 symptoms were found to fluctuate in severity both within and between days, and these severities fluctuated in co-occurrence with each other. The topic is important and interesting. This is a revised version and the authors have taken the suggestions into consideration and the manuscript has improved. The clinical condition of the patients can be a factor impact the post COVID19 symptom and the authors listed comorbidities in the Results section. No major concerns were detected in the manuscript. It would be great if the authors can discuss why MLR models were selected for the analysis.

7. PLOS authors have the option to publish the peer review history of their article (what does this mean?). If published, this will include your full peer review and any attached files.

Reviewer #2: No

Reviewer #3: No

---

## [Author Response · Author response to Decision Letter 1]

17 May 2024

Thank you again for inviting us to revise our manuscript. We have outlined our responses to the reviewers in a separate document labeled 'Responses to Reviewers'.

---

## [Decision Letter · Decision Letter 2]

14 Jun 2024

PONE-D-23-37972R2

Affect and post-COVID-19 symptoms in daily life: An exploratory experience sampling study

PLOS ONE

Dear Dr. Schaap,

Thank you for submitting your manuscript to PLOS ONE. After careful consideration, we have decided that your manuscript does not meet our criteria for publication and must therefore be rejected.

Specifically:

PONE-D-23-37972R1

"Affect and post-COVID-19 symptoms in daily life: An exploratory experience sampling study"

After reviewing the insights provided by the reviewers, it has been determined that this study does not meet the publication standards for the following reasons:

1. The study's results may not be generalizable due to the small sample size, with only 10 participants included in the analysis.

2. Previous publications with larger datasets have already established high correlations among the symptoms, making this study not particularly novel.

3. The study lacks baseline data for the sample and is primarily descriptive in nature.

4. The study does not meet the guideline of at least 10 subjects per factor.

5. The methodology used for the small sample size is questionable.

6. It is advised that the authors address these concerns by collecting more data and analyzing it according to the recommendations provided by the reviewers.

One of the critiques from the reviewer that the author addressed is that the study is beyond its scope. The author mentioned that "in-depth information on all (self-)reported signs and symptoms attributed to post-COVID-19 syndrome is beyond the scope of our study." This statement should be included as a limitation in the manuscript. However, while reading the second revision, this statement was not included. They also need to include a table with the baseline characteristics of the sample and provide a flow diagram of the subjects.

I am sorry that we cannot be more positive on this occasion, but hope that you appreciate the reasons for this decision.

Kind regards,

G. K. Balasubramani

Academic Editor

PLOS ONE

Reviewers' comments:

Reviewer's Responses to Questions

**Comments to the Author**

1. If the authors have adequately addressed your comments raised in a previous round of review and you feel that this manuscript is now acceptable for publication, you may indicate that here to bypass the “Comments to the Author” section, enter your conflict of interest statement in the “Confidential to Editor” section, and submit your "Accept" recommendation.

Reviewer #2: All comments have been addressed

2. Is the manuscript technically sound, and do the data support the conclusions?

Reviewer #2: (No Response)

3. Has the statistical analysis been performed appropriately and rigorously? 

Reviewer #2: (No Response)

4. Have the authors made all data underlying the findings in their manuscript fully available?

Reviewer #2: (No Response)

5. Is the manuscript presented in an intelligible fashion and written in standard English?

Reviewer #2: (No Response)

6. Review Comments to the Author

Reviewer #2: All my concerns are addressed.

The statistics are acceptable.

7. PLOS authors have the option to publish the peer review history of their article (what does this mean?). If published, this will include your full peer review and any attached files.

Reviewer #2: No

- - - - -

---

## [Author Response · Author response to Decision Letter 2]

18 Jul 2024

(Copy from the uploaded Responses to reviewers document:)

Dear editors,

 Herewith, we submit a formal appeal to the rejection of our manuscript, titled Affect and post-COVID-19 symptoms in daily life: An exploratory experience sampling study.

In our opinion, the rejection is based on several mistakes. After submitting the manuscript on 15 November 2023, we were originally invited by the editor (on 4 January) to address some minor revisions to the manuscript and to make our dataset publicly available. After complying accordingly with the remarks of the editor and reviewer, the editor became unavailable. On 3 May, we received another round of feedback from reviewers, which was surprising to us, as we thought the manuscript was all but accepted. Moreover, based on the comments we received, we do not think that these reviewers were able to correctly assess the study on its merits.

Below we respond to the reasons provided point by point:

1. The study's results may not be generalizable due to the small sample size, with only 10 participants included in the analysis.

We agree with the editor that the sample size is small, and we have acknowledged this in our cover letter. Our study is explorative in nature (indicated in the title) as experience sampling methodology (ESM) studies often are. Our aim was to provide some insight into the daily life of people with post-COVID-19 syndrome, and generalisability to wider populations was not our objective. We argue that this is the start of future research in a field that requires more attention (namely the relationship between (psycho)somatic symptoms and affect/emotional well-being in daily life). Moreover, we have indicated the small sample size as a limitation in our discussion. 

2. Previous publications with larger datasets have already established high correlations among the symptoms, making this study not particularly novel.

We also acknowledge this statement. However, this correlational analysis was not an objective of our study, and served to provide context for the main analyses: the fluctuation of post-COVID symptoms in daily life and the association between these symptoms and affect over time. 

3. The study lacks baseline data for the sample and is primarily descriptive in nature.

Baseline characteristics have been provided in our manuscript (page 8), but were other than descriptive not of interest for our objectives, that is, not relevant for our qualitative or statistical assessments. This criticism has not been provided to us by reviewers before.

As mentioned in point 1, we acknowledge that our study is mostly descriptive. To our awareness, PLOS ONE does not restrict accepting descriptive studies, as other ESM studies but also, for example, qualitative studies have been published here before. 

4. The study does not meet the guideline of at least 10 subjects per factor.

As we have outlined in our responses to the reviewers, this guideline is not only arbitrary and strongly debated for linear models, it does not apply to multilevel models such as multilevel linear regression, which require repeated observations. Moreover, the editor has not acknowledged that our analysis is primarily qualitative, and the statistical models support our interpretation of the data. This argument was also provided in our response to the reviewers.

5. The methodology used for the small sample size is questionable.

There are no guidelines for sample sizes in ESM studies, as the number of sampled observations can vary wildly. Key is that the number of observations should be sufficient for the purpose (although here too there are no guidelines for what that entails). With 750 included observations, we think this holds true for an explorative study. We do not test hypotheses, but have set out to explore (co)variation of symptoms and affect qualitatively and statistically.

6. It is advised that the authors address these concerns by collecting more data and analyzing it according to the recommendations provided by the reviewers.

As has been outlined above, we respectfully disagree with the recommendations provided by the reviewers and editor. 

One of the critiques from the reviewer that the author addressed is that the study is beyond its scope. The author mentioned that "in-depth information on all (self-)reported signs and symptoms attributed to post-COVID-19 syndrome is beyond the scope of our study." This statement should be included as a limitation in the manuscript. However, while reading the second revision, this statement was not included. They also need to include a table with the baseline characteristics of the sample and provide a flow diagram of the subjects.

We agree with the editor that we should have acknowledged this as a potential limitation in our discussion. We have added this in the limitation section (p. 15 of the attached revised manuscript). The requirements to include baseline characteristics as a table (and not an in-text description as now is the case) and a flow diagram have not been mentioned before. We could therefore not have included these in our revisions. For the sake of conciseness, we have not added these to the revised manuscript, but are willing to comply if the editor and reviewers think this would improve the manuscript.

Best regards,

Gerko Schaap, Marleen Wensink, Carine Doggen, Job van der Palen, Harald Vonkeman, and Christina Bode

---

## [Editor Report · Decision Letter 3]

15 Oct 2024

Affect and post-COVID-19 symptoms in daily life: An exploratory experience sampling study

PONE-D-23-37972R3

Dear Dr. Schaap,

We’re pleased to inform you that your manuscript has been judged scientifically suitable for publication and will be formally accepted for publication once it meets all outstanding technical requirements.

Kind regards,

Steve Zimmerman, PhD

Senior Editor, PLOS ONE
---

## [Editor Report · Acceptance letter]

16 Oct 2024

PONE-D-23-37972R3 

PLOS ONE

Dear Dr. Schaap, 

I'm pleased to inform you that your manuscript has been deemed suitable for publication in PLOS ONE. Congratulations! Your manuscript is now being handed over to our production team.

Kind regards, 

on behalf of

Dr Steve Zimmerman 

Staff Editor

PLOS ONE